# Emergent CuWO$_4$ Photoanodes for Solar Fuel Production: Recent Progress and Perspectives

**Jin Uk Lee [1], Jin Hyun Kim [2,\*] and Jae Sung Lee [1,\*]**

[1]  School of Energy and Chemical Engineering, Ulsan National Institute of Science and Technology (UNIST), 50 UNIST-gil, Ulsan 44919, Republic of Korea; leejw6588@unist.ac.kr

[2]  Laboratory of Photonics and Interfaces, Institute of Chemical Sciences and Engineering, École Polytechnique Fédérale de Lausanne, 1015 Lausanne, Switzerland

\*  Correspondence: jinhyun.kim@epfl.ch (J.H.K.); jlee1234@unist.ac.kr (J.S.L.)

**Abstract:** Solar fuel production using a photoelectrochemical (PEC) cell is considered as an effective solution to address the climate change caused by CO$_2$ emissions, as well as the ever-growing global demand for energy. Like all other solar energy utilization technologies, the PEC cell requires a light absorber that can efficiently convert photons into charge carriers, which are eventually converted into chemical energy. The light absorber used as a photoelectrode determines the most important factors for PEC technology—efficiency, stability, and the cost of the system. Despite intensive research in the last two decades, there is no ideal material that satisfies all these criteria to the level that makes this technology practical. Thus, further exploration and development of the photoelectode materials are necessary, especially by finding a new promising semiconductor material with a suitable band gap and photoelectronic properties. CuWO$_4$ (n-type, E$_g$ = 2.3 eV) is one of those emerging materials that has favorable intrinsic properties for photo(electro)catalytic water oxidation, yet it has been receiving less attention than it deserves. Nonetheless, valuable pioneering studies have been reported for this material, proving its potential to become a significant option as a photoanode material for PEC cells. Herein, we review recent progress of CuWO$_4$-based photoelectrodes; discuss the material's optoelectronic properties, synthesis methods, and PEC characteristics; and finally provide perspective of its applications as a photoelectrode for PEC solar fuel production.

**Keywords:** solar fuels; photoelectrochemical cell; photoelectrode; copper tungstate; water splitting

## 1. Introduction

The climate change due to rapidly increasing atmospheric CO$_2$ concentration has become an imminent threat to the sustainability of human beings' lives on earth [1]. Solar fuel production is considered an effective strategy to address this urgent global issue, which harvests solar energy and converts and stores it as chemical energy [2]. Three technologies are available for solar fuel production: the photovoltaic cell–electrolyzer (PV-EC) combination is highly efficient but expensive; particulate photocatalysis (PC) is low cost but also low efficiency; the photoelectrochemical (PEC) cell is positioned in the middle of these two technologies, having modest efficiency and cost [3,4]. All the three technologies need to be developed further to create a highly efficient but low cost practical system.

In particular, the PEC device can have a diverse configuration, ranging from a buried junction to a semiconductor-liquid junction (SCLJ) for photovoltage generation [3], and, thus, can utilize a variety of light absorber materials from PV grade materials to stabilize metal oxide semiconductors [4]. Since the long-term stability (>10 years) of aqueous electrolytes is essential for the PEC cells, SCLJ devises are the most staple and general of those used [5]. Also, fundamental studies of the charge transfer mechanism have been focused on SCLJ type PEC cells. For instance, the reaction order for PEC SCLJ water oxidation with Fe$_2$O$_3$ photoanode was found to be in the third order in respect to the

hole concentration with an activation energy ($E_a$) of 60 meV, which is different from those for typical electrochemical water oxidation (the first order, $E_a > 300$ mV) [6]. Such uniqueness makes further study and the development of the SCLJ PEC cell more interesting and worthwhile.

The PEC water splitting starts from the absorption of photons with energy greater than the band gap energy ($E_g$) of the semiconductor (SC) comprising the photoelectrode [7]:

$$SC + h\nu\ (\geq E_g) \rightarrow e_{cb}^- + h_{vb}^+ : \text{Light absorption} \tag{1}$$

Electrons in the conduction band (CB) ($e_{cb}^-$) with an energy level above 0.0 V vs. RHE diffuse to the surface and reduce water to generate hydrogen on a hydrogen evolution co-catalyst (HEC). Holes in the valence band (VB) ($h_{vb}^+$) with an energy level below 1.23 V vs. RHE also diffuse to the surface and oxidize water to generate oxygen, often on an oxygen evolution co-catalyst (OEC). This process leads to the overall water-splitting reaction and produces hydrogen from water using solar energy.

$$4H^+ + 4e_{cb}^- \rightarrow 2H_2 : \text{Hydrogen evolution reaction (HER)} \tag{2}$$

$$2H_2O + 4h_{vb}^+ \rightarrow O_2 + 4H^+ : \text{Oxygen evolution reaction (OER)} \tag{3}$$

$$\text{Overall water splitting: } 2H_2O \rightarrow 2H_2 + O_2,\ \Delta G° = 238\ \text{kJ mol}^{-1} \tag{4}$$

Theoretically, a semiconductor can split water if its $E_g$ is greater than 1.23 eV (water dissociation energy) and its CB and VB edge energy levels are more negative and more positive than the water reduction (0.0 $V_{RHE}$) and oxidation (1.23 $V_{RHE}$) potentials, respectively [8]. In reality, a much larger $E_g$ than 1.23 eV is needed to overcome the overpotentials of the reactions (~0.05 V for HER and ~0.25 V for OER); thus, a total >1.5 V is needed [7,8]. In addition, the rate of reaction is much smaller than the rate of light absorption because of dominant electron–hole recombination, which is the main energy loss process.

General guidelines for selecting metal oxide light absorbers that could be used as photocatalysts for water splitting have often been suggested in previous reviews [7,9,10]. Metal oxide should have adequate energy levels for VBM and CBM, as well as chemical inertness against self-oxidation and a reduction in electrolytes. The VBM is mainly composed of the O 2p orbital of bonding state, while the CBM is often composed of $d^0$ ($Ti^{4+}$, $Zr^{4+}$, $Nb^{5+}$, $Ta^{5+}$, $V^{5+}$, $W^{6+}$, and $Ce^{4+}$) and $d^{10}$ ($Zn^{2+}$, $In^{3+}$, $Ga^{3+}$, $Ge^{4+}$, $Sn^{4+}$, and $Sb^{5+}$) compounds. However, the majority these semiconductors have large band gaps (3.0 eV~3.2 eV) and only absorb UV light. For example, titanates ($TiO_2$, $SrTiO_3$) have proper band alignments for overall water splitting with high quantum yields, but they can only absorb UV light, which limits the solar-to-hydrogen (STH) conversion efficiency to a very low level (the reported highest value is 0.65% [11]).

This generated demand for metal oxide semiconductors that are active under visible light ($E_g < 3.0$ eV). Thus, $WO_3$ ($E_g = 2.8$~2.6 eV) and $Fe_2O_3$ ($E_g = 2.1$ eV) have been extensively studied for generating photocurrent densities ($J_{ph}$) above 4.0 mA/cm$^2$ @ 1.23 $V_{RHE}$ under one sun irradiation (100 mW/cm$^2$) [12].

As such a performance of the binary oxides was not satisfactory, tertiary metal oxides were introduced to form adequate electronic structures with smaller band gaps. One of the most successful cases is $BiVO_4$ with VBM composed of Bi 6s and O 2p, which gives a respectable $E_g$ of 2.4 eV. With a relatively high diffusion length and charge mobility, the state-of-the art $BiVO_4$ photoanode records a $J_{ph}$ above 6.0 mA/cm$^2$ @ 1.23 $V_{RHE}$, which represents the highest among all metal oxide light absorbers [13]. $CuWO_4$, which is one of the photoactive ternary tungstates ($Bi_2WO_6$, which showed PEC water oxidation activity [14], and $ZnWO_4$, which showed pollutant degradation [15] and PEC water oxidation activity [16]), is also a tertiary metal oxide, with CBM composed of W5d and Cu3d and VBM composed of Cu 3d and O 2p, and it could be used as a photocatalyst and a photoanode for water oxidation [17,18]. Although it has a strong indirect band gap property, its smaller

band gap ($E_g$ = 2.3 eV) than that of $WO_3$ ($E_g$ = 2.7 eV) is quite attractive. Hence, here we discuss $CuWO_4$'s potential as an emergent photoanode material for solar fuel production.

## 2. $CuWO_4$ as a Photoanode Material

### 2.1. Crystal Structure of Photoactive $CuWO_4$

The $CuWO_4$ has a unique crystal structure made of distorted wolframite because W ions in $WO_3$ are partially replaced by Cu ions. $W^{6+}$ ions and $Cu^{2+}$ ions are coordinated with $O^{2-}$ ions in octahedral sites (Figure 1a). This structure has a space group of P1$^-$ under low pressure at room temperature, with a = 4.694 A°, b = 5.830 A°, c = 4.877 A°, $\alpha$ = 91.64°, $\beta$ = 92.41°, and $\gamma$ = 82.91°. It is influenced by the Jahn–Teller effect caused by $Cu^{2+}$ ions. This effect leads to distortions in the [$CuO_6$] octahedral clusters, resulting in d orbital splitting and the breaking of the degeneracy of σ-antibonding orbitals. The unpaired electron in the $d_{x2-y2}$ orbital of $Cu^{2+}$ ions, as dictated by the Pauli exclusion principle, creates a mid-gap band state, with greater stabilization observed in Jahn–Teller-elongated $Cu^{2+}$ ions, where the $3d_{z2}$ orbital contains two electrons [19,20].

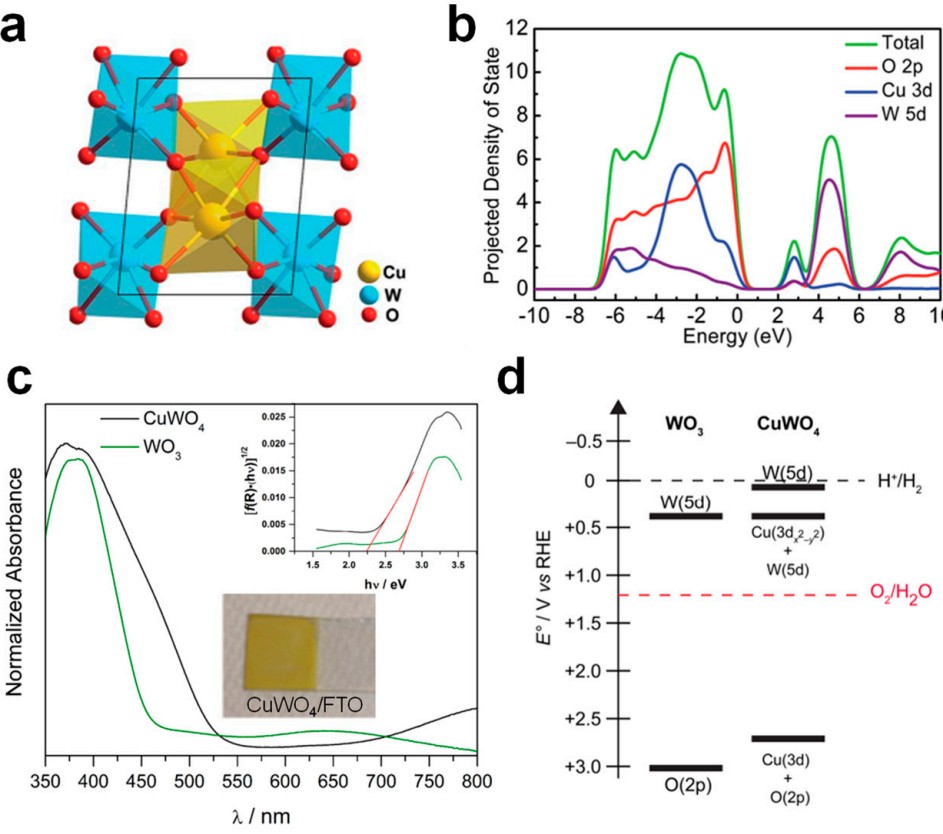

**Figure 1.** (**a**) Crystal and electronic structure of photoactive $CuWO_4$. (**b**) Crystal structure of triclinic wolframite $CuWO_4$ and its calculated projected density of state using hybrid DFT, reproduced from ref. [18]. (**c**) UV−vis spectrum of $WO_3$ and $CuWO_4$ thin film (inset 1: picture image of $CuWO_4$/FTO) (inset 2: Tauc plot of $WO_3$ and $CuWO_4$ with trend line for optical direct band gap), (**d**) experimentally determined electronic structure of $WO_3$ and $CuWO_4$ thin film made via electrodeposition and thermal conversion on FTO, reproduced from ref. [21].

### 2.2. Electronic Structure of Photoactive $CuWO_4$

The electronic structure of $CuWO_4$ was elucidated via various spectroscopies and density functional theory (DFT) calculations. The conduction band is primarily characterized by the contribution from Cu 3d orbitals, with the additional participation of W 5d orbitals. Conversely, the valence band exhibits a strong hybridization between O 2p and Cu 3d orbitals, as illustrated in Figure 1b. Thus, $CuWO_4$ has a suitable band position for

water oxidation, an indirect band gap of 2.3~2.4 eV (Cu 3d → Cu 3d), and a direct band of 2.6~2.7 eV (O 2p → Cu 3d). Its UV-vis spectrum features an absorption onset around 550 nm, reflecting a band gap energy of 2.3 eV (Figure 1c,d). A gradual rather than sharp absorption onset is characteristic of an indirect band gap. Such an indirect band gap tends to limit the light harvesting efficiency of $CuWO_4$. The absorptivity coefficient $\alpha$ is markedly high at 6600 cm$^{-1}$ at 400 nm, but it drops rapidly to 1715 cm$^{-1}$ when the wavelength increases to 500 nm [18,20].

$WO_3$ and $CuWO_4$ share similar electronic structures, though a distinctive feature arises in their indirect band gaps. The $CuWO_4$ possesses an indirect band gap 0.4 eV narrower than that of $WO_3$. This smaller band gap allows $CuWO_4$ to absorb longer-wavelength visible light photons, as shown in Figure 1c. The increased spectral absorption is reflected in the theoretical solar-to-hydrogen (STH) conversion efficiency of $CuWO_4$ higher than 10%. However, the characteristics of the Cu 3d state in $CuWO_4$ result in electron localization, which leads to a constraint in electronic properties, specifically a charge-carrier mobility in the order of ~$6 \times 10^{-3}$ cm$^2$ V$^{-1}$ s$^{-1}$ and a diffusion length restricted to approximately 30 nm for photo-excited electrons in the CB. Such limitations inherently hinder the performance of $CuWO_4$ photoanodes, causing a substantial divergence from their theoretical efficiency. Furthermore, $CuWO_4$ exhibits commendable chemical and thermal stability in neutral electrolytes. This notable stability underscores the material's potential for use in various technological applications [22,23].

## 3. Synthesis Chemistry and Photoelectrochemical Stability

### 3.1. Synthesis of CuWO$_4$ Thin Film

J. E. Yourey et al. [21] initially presented $CuWO_4$ as a promising photoanode material in 2011. The material was synthesized through electrodeposition (ED) using an acidic bath containing a mixture of copper and tungsten. The ED method involved potential sweeps from +0.3 V to −0.5 V, carried out over six cycles. The subsequent annealing of this film was conducted at 500 °C for 2 h. This sample showed 0.2 mA/cm$^2$ at 1.23 V$_{RHE}$ in 0.1 M KPi under one sun illumination. Then, J. C. Hill and K. S. Choi [14] produced $CuWO_4$ with an enhanced surface area by depositing an excess of Cu precursor onto electrochemically deposited porous $WO_3$ electrodes. This approach was found to be adaptable for other tungstate materials. The $CuWO_4$ films were thermally treated at 550 °C in air for 6 h and then immersed in a highly acidic solution to purify the deposited samples. This treatment resulted in an 8-micrometer thick film composed of nanoscale $CuWO_4$ particles, attributed to the presence of the thick underlying $WO_3$ film. It was ascertained that a high-temperature annealing process is vital to produce films that are both impurity-free and highly crystalline. They measured the PEC performances under varying pH conditions—strong acid (0.05 M $H_2SO_4$), neutral (0.1 M phosphate buffer), and weak alkaline conditions (0.1 M borate buffer)—to observe the highest J$_{ph}$ of 0.13 mA/cm$^2$ at 1.23 V$_{RHE}$ under weak alkaline conditions.

A different method of $CuWO_4$ synthesis was introduced by D. Hu et al. [24] using a solid-phase reaction with thermal assistance, circumventing the need for electrochemical procedures. The $WO_3$ nanoflakes (NFs) fabricated through a hydrothermal method served as a sacrificial template for the $CuWO_4$ synthesis. An aqueous solution of $Cu(NO_3)_2$ was meticulously drop-cast onto these $WO_3$ nanoflake arrays and subsequently allowed to dry at a moderate temperature. The detailed fabrication procedure of the $CuWO_4$/FTO photoanode is illustrated in Figure 2a. The transformation of the $WO_3$ and $Cu(NO_3)_2$ mixture into $CuWO_4$ was proposed to proceed through the following reactions:

$$2Cu(NO_3)_2(s) \rightarrow 2CuO(s) + 4NO_2(g) + O_2(g) \tag{5}$$

$$WO_3(s) + CuO(s) \rightarrow CuWO_4(s) \tag{6}$$

Upon the conclusion of this thermal process, a gray–yellow composite film of CuO and $CuWO_4$ was produced. The residual CuO was effectively removed by submerging the

film in a 0.5 M HCl solution for 30 min, yielding a vivid yellow film. The X-ray diffraction (XRD) analysis initially showed a distinct monoclinic $WO_3$ pattern in a film, with specific diffraction peaks at 23.3° and 24.5°, as shown in Figure 2b. However, following a solid-phase reaction at 550 °C for 2 h and the elimination of excess CuO at the surface, the $WO_3$ pattern disappeared completely, being replaced by a clear XRD pattern of triclinic $CuWO_4$. This suggests the complete transformation of $WO_3$ into $CuWO_4$ at a lower temperature of 550 °C. Figure 2c exhibits representative SEM images of the resultant $CuWO_4$ film, exhibiting that the flake nanostructure inherent to the $WO_3$ template is mostly preserved. Notably, the synthesized $CuWO_4$ NFs were slightly thicker than the original $WO_3$ NFs. They claimed that this small alteration in thickness could be attributed to the thermal treatment and the solid-phase interaction between CuO and $WO_3$. Owing to their distinctive morphological and crystalline properties, the $CuWO_4$ NF films demonstrated higher PEC activity in comparison to previously reported $CuWO_4$ synthesized via the ED method. Such a distinctive structure with a vast surface area would minimize hole transport distance from within $CuWO_4$ to its interface and reduce the grain boundary concentration. A comparative study of $CuWO_4$ NF films prepared under diverse annealing durations revealed that the sample annealed at 550 °C for 2 h showed the best performance, registering a photocurrent of 0.32 mA/cm$^2$ at 1.23 $V_{RHE}$ in a 0.1 M NaBi buffer, as shown in Figure 2c. Z. Zhang et al. [25] studied the hydrothermal synthesis of $CuWO_4$ films on FTO substrates using a seeding layer at a moderate temperature. Using a tungsten-rich reaction solution led to a $WO_3$-$CuWO_4$ composite film, which displayed enhanced PEC performance due to improved charge separation.

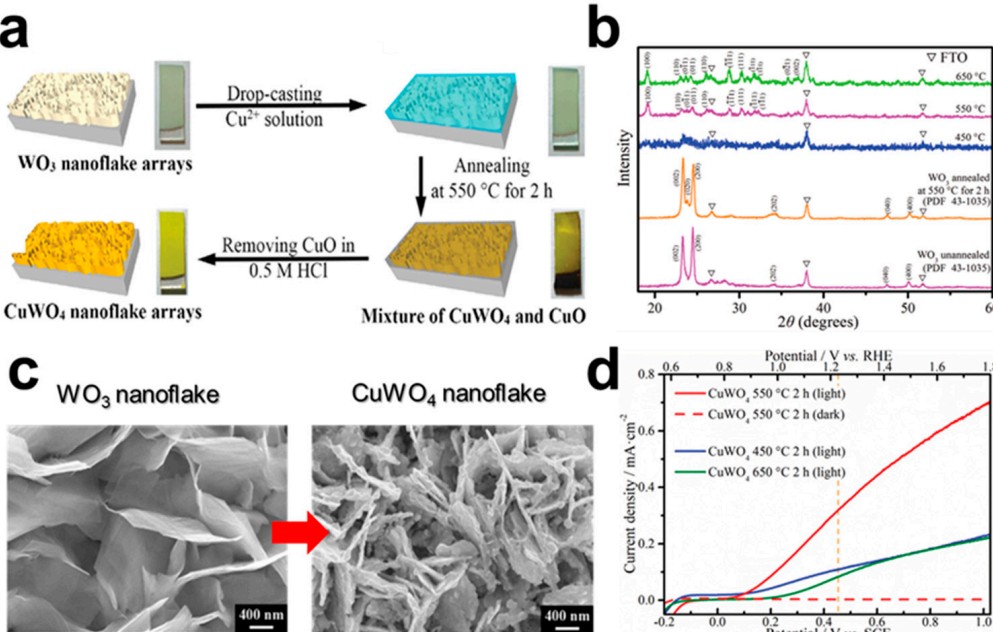

**Figure 2.** Solid-state reaction of $WO_3$ into $CuWO_4$ thin film. (**a**) Schematic illustration of the preparation procedure of the $CuWO_4$ NF array film on an FTO substrate. Optical images represent a $WO_3$ film, a $WO_3$ film drop-cast with $Cu(NO_3)_2$ solution, a mixture film of $CuWO_4$ and CuO, and a $CuWO_4$ film. (**b**) XRD pattern of $CuWO_4$ film annealed at 450 °C, 550 °C, and 650 °C. (**c**) SEM images of the $WO_3$ and $CuWO_4$ nanoflakes prepared at 550 °C for 2 h and (**d**) IV curve of $CuWO_4$ nanoflake photoanode (0.1 M NaBi buffer, pH 9.0) under one simulated sun. Reproduced from ref. [24].

C. M. Tian et al. [18] presented a solution-based synthesis approach that produced thick, high-quality, and polycrystalline $CuWO_4$ films. These films had distinct optoelectronic characteristics and a notable $J_{ph}$ of ~0.5 mA/cm$^2$ at 1.23 $V_{RHE}$. Later, J.U. Lee et al. [26] developed a greenish, transparent $CuWO_4$ polycrystalline film on FTO using a spin coating method, followed by annealing at 550 °C for 2 h. This synthesis protocol

and optical images are shown in Figure 3a,b. The absorption spectrum of the three-layered $CuWO_4$ film in Figure 3c exhibits an absorption edge extended to 510 nm. The Tauc plots reveal indirect and direct band gaps of 2.43 eV and 2.89 eV, respectively, which are similar to the existing literature on polycrystalline $CuWO_4$ films. Moreover, the XRD for the three-layered $CuWO_4$ electrodes in Figure 3d clearly displays the peaks associated with the triclinic $CuWO_4$ phase. The surface and cross-sectional images of the $CuWO_4$ film observed via SEM in Figure 4d exhibit a worm-like grain morphology, forming a porous structure with an approximate thickness of 650 nm and feature sizes of 100–200 nm (Figure 4e) [26]. M.J.d.S. Costa et al. [27] compared various binary metal tungstates ($AWO_4$, $A^{2+}$ = Fe, Cu, Ni, and Co) as photoanode materials synthesized via a sol–gel method using a complexation of metal alkoxides and an esterification/polymerization reaction. Among $AWO_4$ materials, $CuWO_4$ showed the best PEC oxidation performance, recording a $J_{ph}$ of 30 uA/cm$^2$ @ 0.6 Ag/AgCl, while the position of the flat band potential ($E_{fb}$) related to photovoltage was the lowest. X. Duan et al. [28] reported a controllable fabrication method of ultrathin $CuWO_4$ films via an automatic ultrasonic spray pyrolysis method. They extensively explored the effects of varying tungsten sources and film thicknesses on the photoelectrochemical (PEC) performances of the synthesized $CuWO_4$ films. In particular, a $CuWO_4$ film derived from ammonium meta tungstate with a thickness of approximately 2.16 μm demonstrated the best PEC performance (41 μA/cm$^2$ at 1.23 $V_{RHE}$). Then, N. Gaillard et al. [29] reported nanocomposite $CuWO_4$ thin films produced using spray pyrolysis from solutions of copper acetate, ammonium meta tungstate, and MWCNT, resulting in porous, crack-free polycrystalline $CuWO_4$ films with nanoparticle sizes of 10–50 nm. The electrochemical impedance (EIS) tests under one sun illumination showed a 30% reduction in bulk resistance for nanocomposite $CuWO_4$ photoanodes compared to the bare one. The $CuWO_4$ nanocomposite revealed improved a $J_{ph}$ of 0.38 mA/cm$^2$ at 1.63 $V_{RHE}$ in a pH 10 buffer, and MNCNT acted as effective electron collectors throughout the $CuWO_4$ bulk.

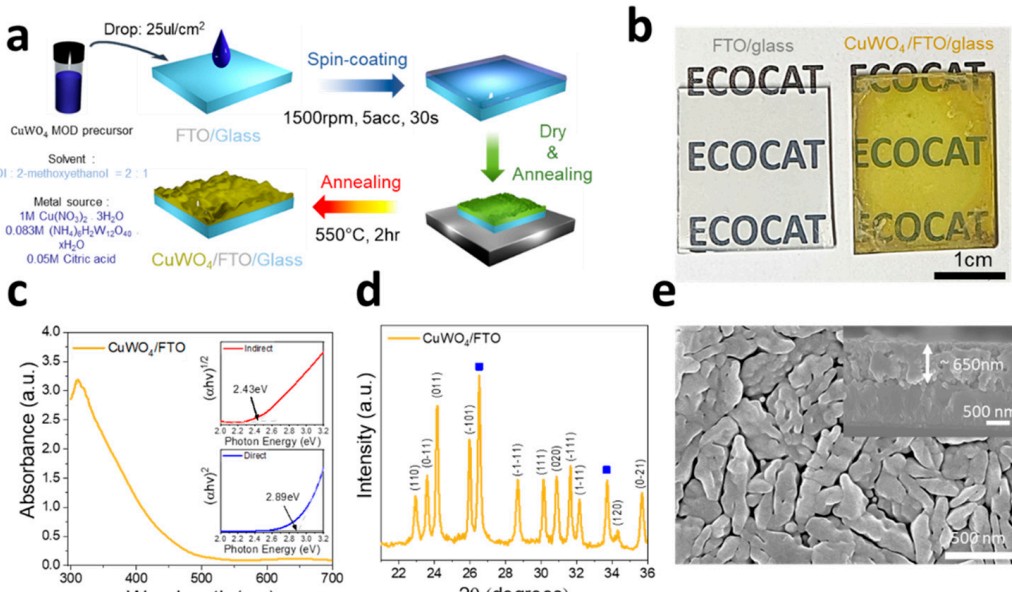

**Figure 3.** Polycrystalline $CuWO_4$ film synthesized via a sol–gel coating method: (**a**) schematics of $CuWO_4$ film coatings on FTO using 1.0 M Cu and W precursors, (**b**) optical images of the films, (**c**) UV-vis spectrum and Tauc plot (insets), (**d**) XRD pattern (blue square indicates pattern of FTO, F doped $SnO_2$ as substrate), and (**e**) a SEM image of $CuWO_4$/FTO. Reproduced from ref. [26].

Meanwhile, Y. Gao et al. [30] fabricated $CuWO_4$ with a Cu:W ratio of 1:1 via the atomic layer deposition method. Subsequent annealing at 600 °C for 30 min facilitated a solid-state reaction between CuO and $WO_3$, resulting in $CuWO_4$ films with elaborately

controlled thicknesses and compositions. This $CuWO_4$ showed 0.11 mA/cm$^2$ at 1.23 $V_{RHE}$ in 0.2 M pH 9 KCl. The chemical vapor deposition method was applied by D. Peeters et al. [23] as another synthesis method that makes it easy to control the stoichiometry of Cu and W in $CuWO_4$ by employing $[Cu(etaoac)_2]$ and $[W(NtBu)_2(dpamd)_2]$ precursors. By manipulating process parameters, they varied Cu-to-W ratios, enabling in-depth studies of the effects of stoichiometry on optical and PEC properties. The $CuWO_4$ of Cu:W = 1:1 showed the best performance ($J_{ph}$ = 0.06 mA cm$^{-2}$) at 1.23 $V_{RHE}$ under frontside one sun illumination.

C. M. Gonzalez et al. [31] proposed pulsed laser deposition (PLD) onto an insulating substrate for synthesizing $CuWO_4$ thin films. They investigated the temperature dependence of electronic conductivity for $CuWO_4$ films across the 100–500 °C temperature range. Although they did not measure the PEC performance, they showed that $CuWO_4$ can be synthesized via PLD. A. Hrubantova et al. [32] made $CuWO_4$ films via reactive high-power impulse magnetron sputtering (HiPIMS) in an argon/oxygen gas mixture using two pieces of magnetron for Cu and W. They observed that both the composition and crystal structure of the as-deposited and post-annealing films depended on the deposition conditions. The samples synthesized on the FTO compromised the $WO_3$ and $CuWO_4$ or $Cu_2WO_4$ phases.

Thus, $CuWO_4$ photoanodes can be synthesized via a variety of methods, but none of them has distinguished itself through outstanding PEC performance. Further exploration of more effective synthesis methods is in order.

### 3.2. Stability of $CuWO_4$

A fundamental requirement for photoelectrodes is their chemical stability in aqueous solutions under light irradiation. The photostability of $CuWO_4$ photoanodes was assessed in comparison to $WO_3$ (Figure 4a,b) in different electrolytes (KPi and KBi buffer, pH 7) (Figure 4c–f). In KBi buffer at pH 7, $CuWO_4$ photoanodes exhibited remarkable stability, maintaining 93% of their initial photocurrent density over a 12-h period. Conversely, in KPi buffer of the same pH, the photocurrent experienced a 50% decline within the initial 4 h and further deteriorated to only the 15% level after a 12-h period. This stark difference implies that the phosphate anion impairs the stability of $CuWO_4$ in aqueous solutions. The $CuWO_4$ electrodes also were tested under higher irradiance conditions to evaluate the degradation of the electrode. The rate of electrode degradation increased in all buffer solutions. The overall trend remained the same, with $CuWO_4$ photoanodes exhibiting better stability in the KBi buffer compared to the KPi buffer.

The $CuWO_4$ photoanodes exhibit a faradic efficiency approaching unity for water oxidation in an equimolar KBi and NaCl solution at pH 7, underscoring their potential utility in this application. In particular, the chronoamperometry (j-t) profiles for 0.1 M KBi and 0.1 M KBi with 100 mM NaCl remain stable under irradiation of both 100 and . In contrast, $WO_3$ begins to dissolve within the initial 30 min of illumination. After a 12-h soaking in KPi electrolyte, the $WO_3$ film displayed a substantial reduction in photocurrent from 0.17 to 0.03 mA/cm$^2$. This enhanced stability of the $CuWO_4$ film arises from a boosted covalency in Cu-O bonds, which hinders the acid−base reaction in $WO_3$ at >pH 5. Moreover, such stability is preserved during water splitting, which is distinctly different to the $WO_3$ case that generates peroxo intermediates during the reaction, thus hastening its degradation [17,21].

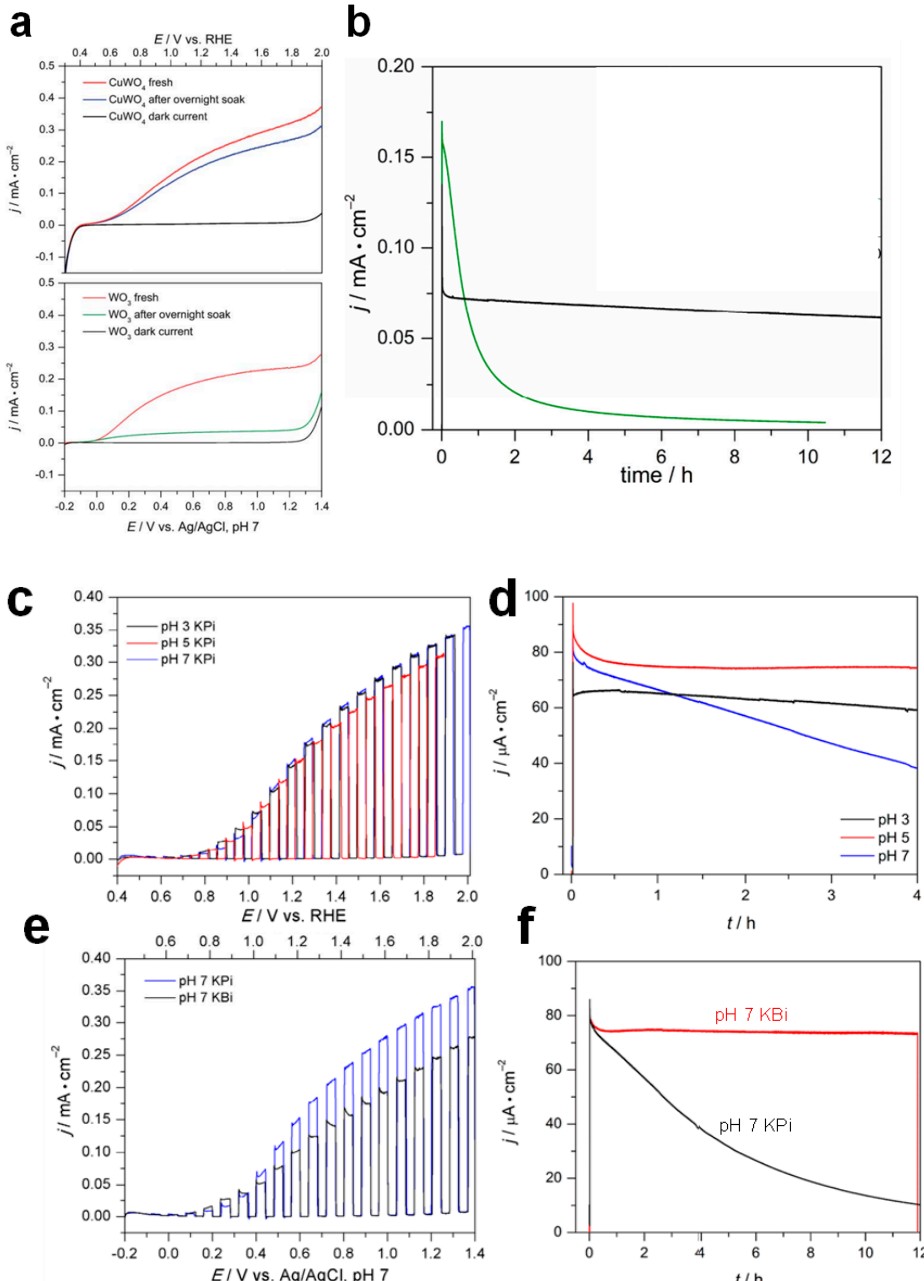

**Figure 4.** Stability of CuWO$_4$ photoanode in various conditions. (**a**) IV curves of CuWO$_4$ and WO$_3$. The blue and green traces show photocurrent density after soaking the electrodes in 0.1 M KPi overnight for CuWO$_4$ and WO$_3$, respectively. The photocurrent densities for both of the fresh photoanodes are shown in red. The black curves represent dark currents. (**b**) Chronoamperometry curves of water electrolysis with CuWO$_4$ (black) and WO$_3$ (green) at 0.5 V vs. Ag/AgCl in a 0.1 M phosphate buffer (pH 7). Reproduced from ref. [21]. CuWO$_4$ photoanode tested in different pHs (**c**,**d**) and electrolytes (**e**,**f**). (**c**,**e**) IV curves and (**d**,**f**) chronoamperometry with an applied potential of 1.23 V$_{RHE}$. All experiments were conducted under simulated AM 1.5G illumination (100 mW/cm$^2$). Reproduced from ref. [33].

## 4. Photoelectrochemical Properties of CuWO$_4$ Photoanodes

As discussed in previous sections, CuWO$_4$ meets important criteria for being a suitable photoanode material to be used in aqueous electrolytes. As an n-type semiconductor, it has an appropriate electronic structure for hole-driven oxidation, with a VBM potential (2.85 V$_{RHE}$) deep enough to overcome the water oxidation redox potential (1.23 V$_{RHE}$)

and a proper band bending at the semiconductor–electrolyte interface for smooth hole transfer [17]. Its inherent stability in near-neutral electrolytes under light irradiation warrants diverse applications involving hole oxidation.

Before the fabrication of a photoelectrode, it is good practice to test the photoactivity of the material in a simpler system, like photovoltaics or a photocatalyst. $CuWO_4$ was found to conduct photocatalytic water oxidation in an electron scavenger ($Ag^+$ or $Fe^{2+}$)-containing solution, thus demonstrating that it is a good candidate material for a photoanode. Thus, Z. Wu et al. demonstrated that powder-type $CuWO_4$ showed n-type conductivity and the formation of a Schottky junction via contact potential difference (CPD) analysis, as well as photocatalytic water oxidation capability using an electron scavenger [34].

Studies of fundamental PEC properties indicated that thin films of $CuWO_4$ could successfully form a SCLJ. Mott–Schottky (MS) plots in various electrolytes confirmed n-type conductivity with a flat-band potential ($E_{fb}$) around 0.5 $V_{RHE}$ and a quasi-Fermi level for the electron ($E_{fn}$) around 0.63 $V_{RHE}$. The majority carrier density ($N_D$) from the slope of the MS plot was ~$10^{-19}$ $cm^{-3}$, which is comparable to that of $BiVO_4$. The photovoltage ($V_{ph}$) of $CuWO_4$ was ~0.6 V without the electrocatalyst. The maximum $V_{ph}$ can potentially reach 0.8 V if there is perfect interaction with $E_{fn}$–$E_{(O_2/H_2O)}$ with no Fermi level pinning. It is slightly lower than that of $BiVO_4$ (0.05 $V_{RHE}$–1.23 $V_{RHE}$ = ~1.2 V) but comparable to that of $Fe_2O_3$ (0.3 $V_{RHE}$–1.23 $V_{RHE}$ = 0.9 V) [26,35,36]. Besides establishing photovoltage at SCLJ, the formation of water oxidation intermediates at SCLJ was also confirmed in a similar manner to the cases of $Fe_2O_3$ and $BiVO_4$ [35].

A physical model for charge-carrier pathways in $CuWO_4$ under illumination is illustrated in Figure 5a—excitation, mid-gap state trapping, and charge–transfer reactions to the solution. The holes originating from VB possess the potential to directly participate in water oxidation or be localized in an intermediate state. The photoexcited electrons in CB move toward the F-doped $SnO_2$ (FTO) substrate, while some of them recombine with holes or migrate to the surface intermediates. The Nyquist plot for $CuWO_4$ demonstrates two independent charge transfer processes at the surface (Figure 5b). The semicircle known as RC1 is linked to non-Faradaic process in the depletion zone, represented as a parallel relationship between $R_{trap}$ and $C_{sc}$, where $R_{trap}$ is the resistance to trapping/detrapping electrons in/out of the mid-gap state, and $C_{sc}$ is the capacitance of the space–charge region. Conversely, RC2 is related to a Faradaic reaction with the solution represented by $R_{ct,mg}$ and $C_{mg}$, where $R_{ct,mg}$ is the resistance to charge transfer at the surface/solution interface (to perform water oxidation) from the mid-gap state, and $C_{mg}$ is the capacitance of the mid-gap state. As the applied voltage increases, the resistance associated with water oxidation ($R_{ct,mg}$) decreases, while the resistance associated with electron trapping ($R_{trap}$) increases. This behavior suppresses the recombination of photo-induced electrons at the mid-gap and facilitates electron transfer to the charge collector, initiating OER at 0.80 V. The capacitance of the space–charge region ($C_{SC}$) remains constant at potentials exceeding 0.86 V, indicating the continuous flow of electrons to the charge collector. In contrast, the capacitance of the mid-gap state ($C_{mg}$) continues to increase after 0.8 V, representing the charging and discharging of this state as water oxidation proceeds. The $CuWO_4$ showed a remarkable photocurrent increase and unpinning of the Fermi level at 1.06 V. This observation indicates that as the reaction proceeds, electrons from the solution populate the mid-gap state, and the increase in $C_{mg}$ suggests that hole transfer from the valence band to the mid-gap state becomes rate limiting [37].

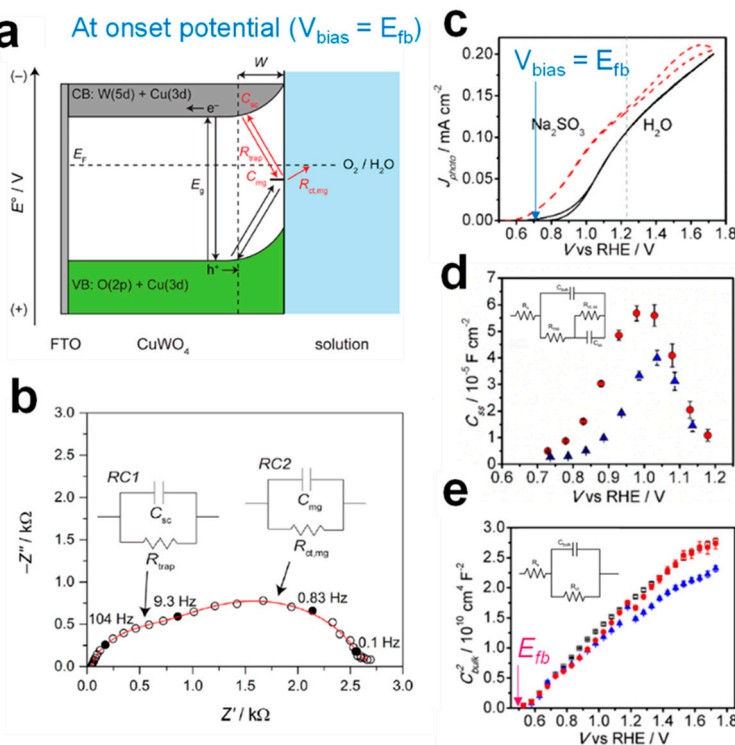

**Figure 5.** Formation of semiconductor–liquid junction for $CuWO_4$ in an aqueous electrolyte. (**a**) Proposed physical model for charge-carrier pathways in $CuWO_4$: excitation, mid-gap state trapping, and charge–transfer reactions to solution. W represents the depletion width, and the red arrows indicate charge–transfer processes. (**b**) Nyquist plot for EIS data measured in 0.5 M KBi and 0.2 M KCl at pH 7, under one sun illumination, and 0.96 $V_{RHE}$. Solid circles in Nyquist plot is marked with frequency of measurement applied to it. Reproduced from ref. [37]. (**c**) IV curves of $CuWO_4$/FTO in pH 9.0 KBi buffer under one sun illumination. (**d**) $C_{ss}$ obtained via fitting EIS under 0.1 sun (blue triangles) and 1 sun (red circles) illumination in $H_2O$ electrolyte, fitted with equivalent circuit for EIS data with two capacitive features (inset). (**e**) Mott–Schottky plots of $CuWO_4$ immersed in $H_2O$ electrolyte, measured in the dark (open squares), under 0.1 sun (blue triangles) and 1 sun (red circles) illumination, fitted with a Randel circuit (inset). Reproduced from ref. [35].

The onset potentials of $CuWO_4$ for the PEC oxidation of water and $Na_2SO_3$ (a hole scavenger) were found to be around 0.85 $V_{RHE}$ and 0.6 $V_{RHE}$, respectively (Figure 5c). The observed shift in onset potential between the water and $Na_2SO_3$ oxidation processes is potentially due to the suboptimal hole collection during water oxidation at the $CuWO_4$ interface. The EIS analysis was conducted for $CuWO_4$ in $H_2O$ and $Na_2SO_3$ solution to explain the observation. The derived surface state capacitance ($C_{ss}$) values diminished as the light intensity was reduced from 1 sun to 0.1 sun (Figure 5d). The peak $C_{ss}$ at ~1.0 $V_{RHE}$ is close to its onset potential for water oxidation, while IV curves indicated that water oxidation intermediates form at ~0.8 $V_{RHE}$. Furthermore, the bulk capacitance ($C_{bulk}$) values decreased with the applied voltage, but they were essentially invariant at a given potential under different light intensities. Using these $C_{bulk}$ values, MS plots were formulated in Figure 5e, which gave $E_{fb}$ and $N_D$ values comparable to the literature's values, as described above [35].

Intensity modulated photocurrent spectroscopy (IMPS) was employed to quantify hole transfer kinetics at SCJL [35,38]. A large impedance of hole transfer at SCLJ turned out to be the major huddle of $CuWO_4$ for water oxidation, just like other known metal oxide semiconductors.

## 5. Modification Strategies of CuWO₄ Photoanodes

Despite having only recently emerged, the PEC water oxidation performance of CuWO₄ photoanodes has steadily improved through various modification strategies, as summarized in Table 1. These strategies include n-type doping, solid solution, heterojunction, electrocatalyst, and post-treatment, and each of these strategies is discussed in detail in this section.

**Table 1.** Summary of progress made for CuWO₄ as a photoanode material.

| Strategies | Reported Substances | Typical Effects |
|---|---|---|
| Synthesis method | Electrodeposition [14,21]<br>Thermal conversion [39,40]<br>Sol–gel method [26,29,35]<br>Hydrothermal [25]<br>ALD [30]<br>CVD [23]<br>Ultrasonic spray pyrolysis [28]<br>Impulse magnetron co-sputtering [32]<br>PLD [31] | Electrodeposition ($J_{ph}$ 0.2 mA/cm² at 1.23 $V_{RHE}$) [21]<br><br>Thermal conversion ($J_{ph}$ 0.33 mA/cm² at 1.23 $V_{RHE}$) [24]<br>Sol–gel ($J_{ph}$ 0.5mA/cm² at 1.23 $V_{RHE}$, 1.0 mA/cm² at 1.23 $V_{RHE}$ for SA oxidation) [18]<br>Sol–gel ($J_{ph}$ 0.07mA/cm² at 1.23 $V_{RHE}$, 0.15 mA/cm² at 1.23 $V_{RHE}$ for SA oxidation) [26] |
| Doping | $Fe^{3+}$ [41]<br>F [42]<br>$Y^{3+}$ [43]<br>$Mo^{6+}$ [44] | 0.3% Fe:CuWO₄/FTO, ~1.5 times greater $\eta_{bulk}$ ($J_{ph}$ 0.5 mA/cm² at 1.23 $V_{RHE}$ for SA oxidation) [41] |
| Mo solid solution | $Mo^{6+}$ [45–48] | Red shift of photoresponse (from 550 nm to 600 nm) [45]<br>CuW$_{0.35}$Mo$_{0.65}$O₄/FTO ($J_{ph}$ 1.0 mA/cm² at 1.23 $V_{RHE}$ for SA oxidation) [46] |
| Heterojunction and electron transfer layer | WO₃ (flat) [49]<br>WO₃ (nanorod) [39]<br>WO₃ (urchin like) [50]<br>SnO₂ [26]<br>BiVO₄ [51] | CuWO₄/flat WO₃/FTO. ~4 times increment ($J_{ph}$ 0.55 mA/cm² at 1.23 $V_{RHE}$) [49] |
| Electrocatalyst | Co-Pi [26,52,53]<br>Co₃O₄ [18]<br>FeCoO$_x$ [54]<br>MnPO₄ [55]<br>NiWO₄ [56]<br>NiFeO$_x$ [57]<br>P-type sulfide (MoS₂, NbS₂, NiS$_x$) [44]<br>MnNCN [58]<br>Ni-Pi [59]<br>Ag [60]<br>IrCo-Pi [61] | Co-Pi/CuWO₄/FTO, 30% increment ($J_{ph}$ 0.4 mA/cm² at 0.6 $V_{Ag/AgCl}$) [52] |
| Post-treatment | H₂ treatment [62] | H₂ treatment (300 °C), 3-time increment ($J_{ph}$ 0.6 mA/cm² at 1.01 $V_{Ag/AgCl}$) [62] |

SA: sacrificial agent. $\eta_{bulk}$ = bulk charge separation efficiency.

### 5.1. Extrinsic/Intrinsic Defect Engineering via Doping

The extrinsic defect engineering conducted via the doping of foreign elements can promote the electron conductivity of CuWO₄ by generating additional charge carriers, which can be characterized using the MS equation.

$$\frac{1}{C^2} = \frac{2\left(V - E_{fb} - \frac{kT}{e}\right)}{e\varepsilon\varepsilon_0 N_D A^2} \tag{7}$$

where $C$ = the capacitance of the photoanode (metal oxide + electrolyte double layer, etc.), $e$ = the charge of the electron, $\varepsilon$ = the dielectric constant of CuWO₄, $\varepsilon_0$ = the permittivity of

the vacuum, $V$ = the applied bias (vs. RHE), $E_{fb}$ = the flat band potential (vs. RHE), $k$ = the Boltzmann constant, $N_D$ = the donor density of the n-type semiconductor (cm$^{-3}$), $A$ = the surface area of photoanode, and $T$ = the temperature (K). There are two kinds of doping methods: one uses a non-isovalent dopant, and the other uses an isovalent dopant. The non-isovalent dopants can enhance electron conductivity by elevating the charge-carrier density of $CuWO_4$, while the isovalent doping also boosts conductivity by improving the charge migration, although it cannot enhance the charge density. The dopant atoms can substitute into $Cu^{2+}$ or $W^{6+}$ octahedral sites, and their efficacy is determined by their suitable oxidation state, ionic size, and electronic configuration.

Fe doping was first applied to $CuWO_4$ as extrinsic doping via spray pyrolysis using a $FeSO_4$ precursor [41]. The iron was selected as a suitable dopant due to its prevalent non-toxic characteristics, its similarity in terms of the ionic radii of $Fe^{3+}$ and $Fe^{2+}$ to the target $Cu^{2+}$ ion, and its good compatibility with the crystalline structure of $CuWO_4$. The reaction (8) expresses the defect sites formed by $Fe^{3+}$ substituting into the $CuWO_4$ lattice, as delineated by the Kröger–Vink notation:

$$Fe_2O_3 + 2WO_3 \rightarrow 2W_W^x + 8O_O^x + \frac{1}{2}O_2(g) + 2Fe_{Cu}^{\bullet} + 2e^- \tag{8}$$

The $CuWO_4$ doped with 0.3% Fe exhibited a 50% increase in photocurrent density compared to the undoped one at 1.23 $V_{RHE}$ in 0.1 M KPi with an $H_2O_2$ hole scavenger. Furthermore, the charge separation efficiency at 1.23 $V_{RHE}$ was 50% higher for the doped sample.

Yttrium was suggested as another extrinsic dopant by S. G. Poggini et al. [43]. Y-doping was conducted via a dip coating method. The Y-doping into $CuWO_4$ thin film led to a minor blue shift in the band gap from that of pristine $CuWO_4$ (2.30 eV) due to a rise in the conduction band position, as supported by DFT calculations. An optimized 5% Y-doped sample showed a photocurrent density 92.5% higher than that of an undoped sample at 1.3 $V_{RHE}$.

Another effective doping method for the $CuWO_4$ crystal lattice is to generate oxygen vacancies ($V_{\ddot{O}}$) through $H_2$ treatment, $N_2$ treatment, or F doping. The $H_2$ treatment increases the charge-carrier density through the partial reduction of the metal oxide:

$$\text{Intrinsic doping}: O_O^x + H_2(g) \rightarrow H_2O(g) + V_{\ddot{O}} + 2e^- \tag{9}$$

Y. Tang et al. [62] conducted $H_2$ treatment for $CuWO_4$ to induce oxygen deficiency in its crystal lattice (Figure 6a,b). The $H_2$-treated $CuWO_4$ film showed a noticeable color change from bright yellow to dark yellow as the annealing time increased because $V_{\ddot{O}}$ can occur at a color center located in the band gap. This color change hints at a possible alteration in the $CuWO_4$ band gap caused by $H_2$ treatment. The UV–visible absorption spectra of bare and modified $CuWO_4$ samples shown in Figure 7a reflect this color change. The 30-min treatment gave the best performance, showing a negative shift of overpotential from 0.35 V to 0.05 V vs. Ag/AgCl, as well as an increased donor density in MS analysis compared to bare $CuWO_4$ (Figure 7b).

The use of nitrogen post-treatment to enhance the PEC performance of $CuWO_4$ by forming oxygen vacancy was suggested by Z. Ma et al. [63]. A nitrogen atmosphere can provide a moderate reductive environment for $CuWO_4$, which neither collapses the crystal structure nor creates a color center in the band gap. As a result, there is no color change in the $CuWO_4$ films. The $N_2$-treated $CuWO_4$ sample annealed at 623 K showed 80 $\mu$A/cm$^2$ @ 1.23 $V_{RHE}$ in 0.1 M KPi under one sun illumination.

C. Li and P. Diao [42] introduced a straightforward method for fluorine doping, where $F^-$ anions were substituted into the crystal lattice of $CuWO_4$ nanoflakes, replacing oxygen atoms and acting as potent electron donors to increase electron density. By modulating the volume of the F precursor solution, the dopant concentration was precisely controlled. The F-doped $CuWO_4$ nanoflakes demonstrated significantly enhanced PEC performance coupled with robust stability in PEC oxygen evolution reactions. The optimized 2.5%

F-doping enhanced the photocurrent density of CuWO$_4$ from 0.32 to 0.57 mA/cm$^2$ at 1.23 V$_{RHE}$.

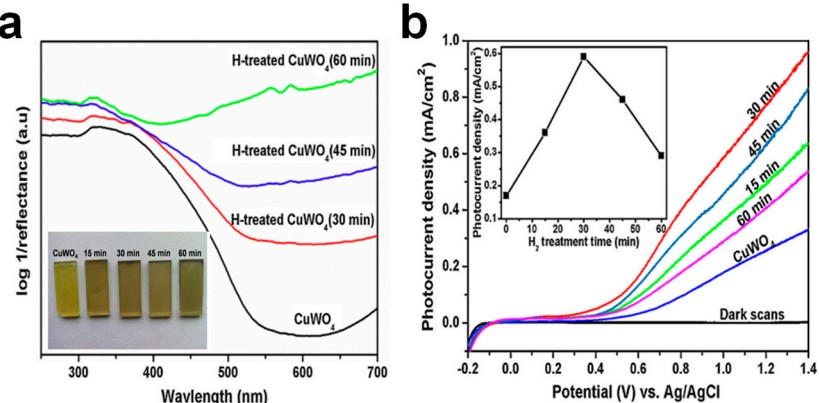

**Figure 6.** H$_2$ gas treatment for partial reduction of CuWO$_4$ photoanode. (**a**) UV-visible absorption spectra of CuWO$_4$ film with H$_2$ gas treatment (300 °C, H$_2$ 5%, Ar 95% flow) with different treatment durations. Inset is an image of a CuWO$_4$ film with H$_2$ gas treatment. (**b**) IV curves of CuWO$_4$ with H$_2$ gas treatment (0.1 M Na$_2$SO$_4$, under simulated one sun). Inset represents the photocurrent densities at an applied bias of 1.01 V$_{Ag/AgCl}$. Reproduced from ref. [62].

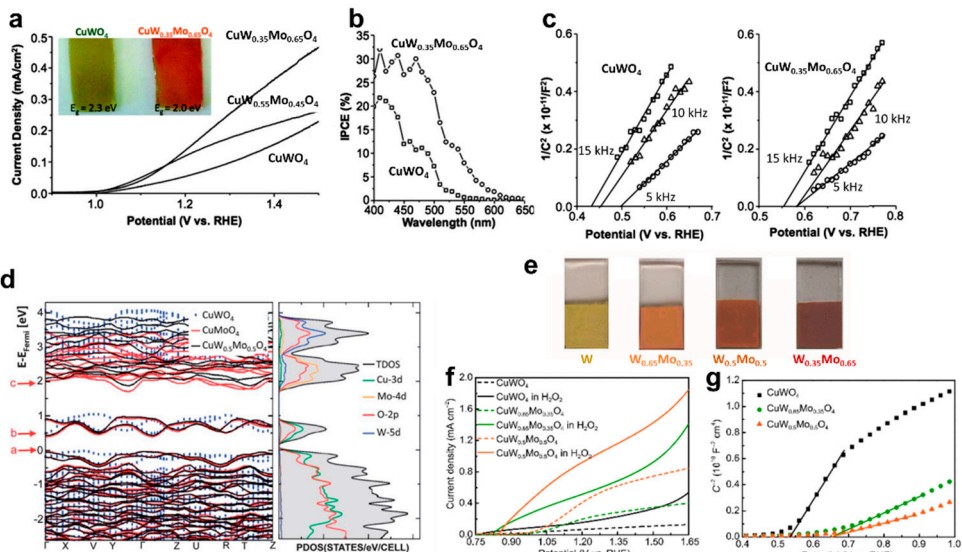

**Figure 7.** CuW$_{1-x}$Mo$_x$O$_4$ solid solution for photoanode. (**a**) IV curves of CuWO$_4$ and solid solutions (W/Mo = 0.5/0.5 and 0.35/0.65) made via the electrodeposition (ED) of the precursor and thermal conversion (0.1 M KPi, pH 7.0, under simulated one sun). (**b**) IPCE under applied voltage of 1.61 V$_{RHE}$, (**c**) Mott–Schottky plots of CuWO$_4$ and CuW$_{0.35}$Mo$_{0.65}$O$_4$ obtained in 0.1 M KPi (pH 7), and (**d**) band structures of CuWO$_4$, Type-III CuMoO$_4$, and CuW$_{0.5}$Mo$_{0.5}$O$_4$ solid solutions, aligned by Mo (W) 4s, 4p (5s, 5p) core states (left) and projected density of states (PDOS) for a CuW$_{0.5}$Mo$_{0.5}$O$_4$ solid solution (right), obtained from DFT calculations (a: the Cu-O state, b: the Cu-O state with underestimated the local spin density approximation (LDA + U), c: the hybrid W/Mo 5d/4d and O 2p states). Reproduced from ref. [45]. (**e**) Images of CuW$_{1-x}$Mo$_x$O$_4$ film made via spray pyrolysis. (**f**) IV curves (0.1 M KPi pH 7.0, under simulated one sun, 0.1 mol% H$_2$O$_2$ as hole scavenger). (**g**) Mott–Schottky plots of CuW$_{1-x}$Mo$_x$O$_4$ film. Reproduced from ref. [46].

## 5.2. Formation of Solid Solution with Mo

Of the modification strategies developed to improve the PEC performance of CuWO$_4$ photoanodes, solid solution with Mo was the most effective and innovative [45,46,48].

The $Mo^{6+}$ ion can easily replace $W^{6+}$ in the $CuWO_4$ crystal structure because they have similar ionic radii and the same valance. The incorporation of Mo reduces the band gap and augments the light harvesting in the visible light region. The reduced band gap for $CuW_{1-X}Mo_XO_4$ is caused by the shift of the CBM to a low energy as Mo orbitals are mixed with Cu and W orbitals. This shift can push $E_{fb}$ below the water reduction potential and make the overall water splitting impossible, but the PEC water oxidation efficiency improves significantly owing to extended light harvesting.

J. C. Hill et al. [45] demonstrated, for the first time, $CuW_{1-X}Mo_XO_4$ photoelectrode prepared using ED and thermal conversion methods. They identified an optimal photoelectrode of $CuW_{0.35}Mo_{0.65}O_4$ with a reddish color that showed the highest PEC performance (Figure 7a). To ascertain that the highest photocurrent comes from extended light harvesting, the incident photon-to-current conversion efficiency (IPCE) of $CuWO_4$ and $CuW_{0.35}Mo_{0.65}O_4$ are presented in Figure 7b. The photocurrent onset for $CuWO_4$ was 560 nm, whereas it shifted to 620 nm for $CuW_{0.35}Mo_{0.65}O_4$. This demonstrates that the increased photon absorption between 560 and 620 nm was directly translated into elevated photocurrents. In addition, $CuW_{0.35}Mo_{0.65}O_4$ showed substantially superior IPCEs across the entire visible light region. The MS plots of $CuWO_4$ and $CuW_{0.35}Mo_{0.65}O_4$ were displayed to discern their $E_{fb}$ values (Figure 7c). Both samples exhibit marginal frequency-dependent variations, but on an aggregate scale, $CuW_{0.35}Mo_{0.65}O_4$ possesses a more positive $E_{fb}$ than $CuWO_4$. This implies that the augmented photocurrents of $CuW_{0.35}Mo_{0.65}O_4$ were caused by an increase in light absorption rather than $E_{fb}$. Moreover, the slopes of the MS plots for both samples appear similar at each frequency, suggesting that the $E_{fb}$ was shifted by the CBM change rather than the variation in charge-carrier density. The band structures of $CuWO_4$, Type-III $CuMoO_4$, and $CuW_{0.5}Mo_{0.5}O_4$ solid solutions were calculated *via* DFT. The VBMs of all samples predominantly comprise hybridized Cu 3d and O 2p states. But the CBMs of $CuW_{0.5}Mo_{0.5}O_4$ and Type-III $CuMoO_4$ are mainly composed of hybridized Mo 4d and O 2p states, while the CBM of $CuWO_4$ consists of hybridized W 5d and O 2p states, as shown in Figure 7d.

Meanwhile, $CuW_{1-X}Mo_XO_4$ was synthesized via spray pyrolysis by Q. Liang et al. [46], who simply adjusted the ratio by adding an Mo source instead of a W source. As the Mo ratio increased, the color of $CuW_{1-X}Mo_XO_4$ changed to reddish, and when the Mo ratio was higher than 35%, $Cu_3Mo_2O_9$ impurity began to appear. Fortunately, the number of impurities was small up to 50% Mo content but increased rapidly beyond that level. These impurities can create a color center within the band and darken the color of $CuW_{1-X}Mo_XO_4$ samples (Figure 7e). The PEC performances of $CuW_{1-X}Mo_XO_4$ samples were measured in 0.1 M phosphate buffer with or without $H_2O_2$ under simulated one sun, and $CuW_{0.5}Mo_{0.5}O_4$ showed the best performance in both electrolytes. In water oxidation, it generated ~0.47 mA/cm$^2$ at 1.23 $V_{RHE}$ under one sun illumination, which was ~7 times higher than that of bare $CuWO_4$. In $H_2O_2$ oxidation, $CuW_{0.5}Mo_{0.5}O_4$ recorded ~0.95 mA/cm$^2$ at 1.23 $V_{RHE}$, and its onset potential shifted by 80 mV in a cathodic direction due to the enhanced reaction kinetics (Figure 7f). The $E_{fb}$ and charge density ($N_D$) of $CuW_{1-X}Mo_XO_4$ were calculated from MS plots, as shown in Figure 7g. $CuW_{0.65}Mo_{0.35}O_4$ and $CuW_{0.5}Mo_{0.5}O_4$ showed more positive $E_{fb}$ than $CuWO_4$, while $N_D$ values estimated from the slopes of the MS plots were higher for bare $CuWO_4$. The authors claimed that $CuW_{0.5}Mo_{0.5}O_4$ had enhanced conductivity and mitigated charge recombination.

A. Polo et al. [48] produced a $CuW_{0.5}Mo_{0.5}O_4$ photoelectrode by stacking layers using a spin coating method. They found that a three-layered sample was optimal, and using four or more layered samples led to electron mobility issues, reducing the performance. The three-layered $CuW_{0.5}Mo_{0.5}O_4$ generated a photocurrent of ~0.25 mA/cm$^2$ at 1.23 $V_{RHE}$ in 0.1 M $K_3BO_3$, which was remarkably higher than $CuWO_4$ prepared via a similar method. Utilizing a similar spin coating method, K. Wang et al. [64] achieved a gradient distribution of Mo within the $CuWO_4$ film while doping the surface layer with Ni. The Mo incorporation caused the CBM to shift to a higher energy level, while the Ni doping induced the VBM to shift to a slightly lower energy, promoting efficient charge separation and alleviating

charge recombination. As a result, the fully modified $CuWMoO_4$ revealed 0.85 mA/cm$^2$ @ 1.23 $V_{RHE}$. Lastly, J. Yang et al. [47] made $CuW_{1-X}Mo_XO_4$ nanoflake films using hydrothermal and thermal solid-state reactions. They first synthesized Mo-WO$_3$ nanoflakes via a solid-state reaction, with desired amounts of Mo and W placed on the WO$_3$ seed layer. Then, a Cu precursor was deposited on it and heated to obtain a $CuW_{1-X}Mo_XO_4$ NFs film. The $CuW_{0.68}Mo_{0.32}O_4$ exhibited the best performance of ~0.62 mA/cm$^2$ at 1.23 $V_{RHE}$ in 0.1 M sodium phosphate buffer solution.

### 5.3. Heterojunction Electron Transfer Layer

An effective heterojunction electron transfer layer (ETL) should possess the following characteristics: (i) enhanced electron transfer properties compared to $CuWO_4$, (ii) a type II staggered band alignment facilitating spontaneous electron transfer while blocking hole transfer, and (iii) a multidimensional high-aspect-ratio structure accompanied by porosity, enabling significant $CuWO_4$ loading while maintaining a modest thickness [65]. WO$_3$ has been commonly used as an ETL to improve the bulk charge separation efficiency for many photoanodes, such as Fe$_2$O$_3$ [66] and BiVO$_4$ [67], because it has an appropriate band gap (~2.7 eV), a high electron mobility (~10 cm$^2$V$^{-1}$S$^{-1}$), and a long hole diffusion length (~800 nm) [68]. Moreover, the synthesis method of $CuWO_4$ from WO$_3$ is well known, and $CuWO_4$/WO$_3$ samples of various structures have also been reported.

D. Wang et al. [49] synthesized $CuWO_4$/WO$_3$ porous thin films through magnetron sputtering, employing a polymer templating approach. Utilizing a dip coating process, a Cu precursor with polymer surfactant was deposited on a WO$_3$ film. Then, $CuWO_4$/WO$_3$ was made via annealing at 550 °C for 2 h (Figure 8a). Figure 8b shows a cross-sectional SEM image of the $CuWO_4$/WO$_3$ composite, which clearly reveals respective WO$_3$ and $CuWO_4$ layers. The WO$_3$ seed layer became a little thinner as $CuWO_4$ of a thickness of 600 nm was formed. The $CuWO_4$/WO$_3$ composite exhibited a remarkably higher PEC performance of ~0.45 mA/cm$^2$ at 1.2 $V_{RHE}$ compared to either the WO$_3$ or $CuWO_4$ photoanodes, as shown in Figure 8c. The IPCE for $CuWO_4$/WO$_3$ thin film at 1.2 $V_{RHE}$, as shown in Figure 8d, was much higher than those of the WO$_3$ and $CuWO_4$ films. Although $CuWO_4$ can absorb photons up to 540 nm, both samples revealed very low IPCEs between 470 and 540 nm, indicating that the charge carriers generated from $CuWO_4$ in this 470~540 nm range contribute little to the photocurrent generation. The CBs of $CuWO_4$ and WO$_3$ are positioned at +0.2 eV and +0.4 eV, respectively, whereas the VBs of $CuWO_4$ and WO$_3$ are positioned at +2.4 eV and +3.0 eV, respectively. When they form a $CuWO_4$/WO$_3$ heterojunction, the photo-induced holes from WO$_3$ migrate to $CuWO_4$, while electrons travel in the opposite direction to the FTO charge collector, as shown in Figure 8e.

T. Wang et al. [50] produced urchin-like nanoarray $CuWO_4$/WO$_3$ films through a one-step hydrothermal reaction by controlling the reaction time. The optimized $CuWO_4$/WO$_3$ nanoarray film demonstrated an onset potential of 0.6 $V_{RHE}$ and a photocurrent density of 0.48 mA/cm$^2$ at 1.23 $V_{RHE}$ in pH 7 solution. An enhanced PEC performance of the film can be attributed to its distinctive urchin-like nanoarray structure that provides a large surface area, in addition to facile charge separation by forming an effective $CuWO_4$/WO$_3$ heterojunction. I. Rodríguez-Gutiérrez et al. also synthesized $CuWO_4$/WO$_3$ photoelectrodes from WO$_3$ nanorod seed [39]. Through various electrochemical analyses, it was shown that WO$_3$ was partially or completely converted to $CuWO_4$, depending on the annealing temperature. It was found that only $CuWO_4$ was present in the sample that was annealed at temperatures of 650 °C or higher. However, the highest efficiency (~0.3 mA/cm$^2$ @ 1.23 $V_{RHE}$ in 0.1 M phosphate buffer) was observed for the sample annealed at 450 °C, which contained a significant amount of WO$_3$. In addition, $CuWO_4$ served as a protective layer for the WO$_3$ material that is not generally stable in neutral aqueous solutions.

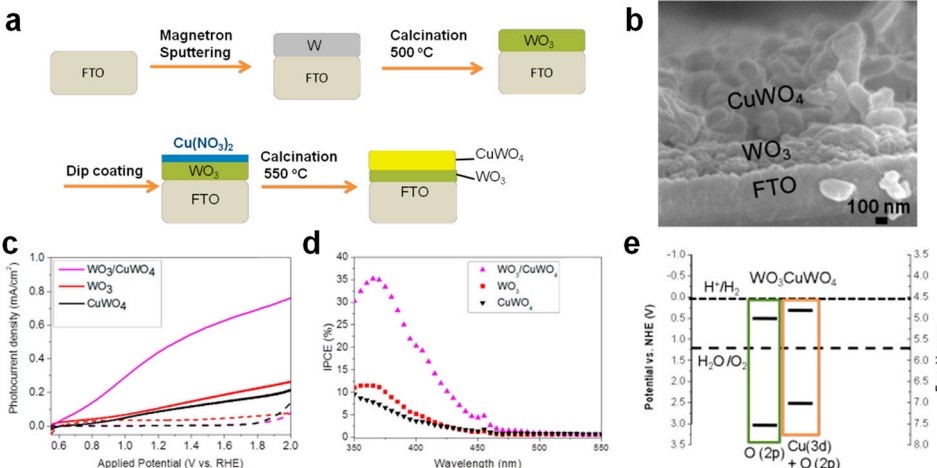

**Figure 8.** Effects of $CuWO_4/WO_3$ type II heterojunction. (**a**) Schematics of the formation of a $CuWO_4/WO_3$ heterojunction film and (**b**) its SEM image. (**c**) IV curves (solid line: under illumination, dotted line: under dark) and (**d**) IPCE at an applied bias of 1.20 $V_{RHE}$ for $CuWO_4$, $WO_3$, and $CuWO_4/WO_3$ photoanodes in 0.5 M $Na_2SO_4$ under one sun. (**e**) Tentative energy diagram of the $CuWO_4/WO_3$ type II heterojunction based on UV-vis spectra and the Mott–Schottky plot. Reproduced from ref. [49].

Meanwhile, another common ETL material, namely $SnO_2$, was also applied to $CuWO_4$ film, which was synthesized via a spin coating method [26]. Although the CBM of $SnO_2$ is located at a higher energy than that of $CuWO_4$, its high electronic conductivity allows electrons to easily move to $SnO_2$, and the VBM of $SnO_2$ prevents the hole transfer from $CuWO_4$. As a result, the $CuWO_4/SnO_2$ sample showed a photocurrent of ~0.1 mA/cm$^2$ (at 1.23 $V_{RHE}$ in 0.1 M KPi) and a bulk charge separation efficiency twice as high as that of bare at 1.23 $V_{RHE}$. The IPCE of $CuWO_4/SnO_2$ showed a diminished disparity in front/back side illumination conditions compared to bare $CuWO_4$. This suggests a more effective electron transfer for the $CuWO_4/SnO_2$ via the hole-blocking effect. In addition, $SnO_2$ ETL can inhibit the electron–hole recombination at the interface of the light absorber and charge collector.

### 5.4. Surface Modification via Co-Catalyst Loading

The surface modifications of $CuWO_4$ usually target the enhancement of charge transfer from a semiconductor to an electrolyte and stability under sun illumination. A simple but most effective way to modify the surface properties is to deposit an electrocatalyst on the semiconductor's surface. The electrocatalyst (termed co-catalyst on photoelectrode) plays multiple roles, such as (i) reducing the activation energy required for water oxidation, (ii) passivating the surface state or defects on the semiconductor, and (iii) storing photogenerated holes to promote a water oxidation reaction that requires multi-hole transfer [69]. The last two functions distinguish the co-catalyst on the photoanode from the typical electrocatalyst using only the first function.

A cobalt phosphate (Co-pi) is the best-known co-catalyst for oxygen evolution reaction (OER), as it is highly effective with various photoanodes. Figure 9a shows the band alignment and working mechanism when Co-Pi is loaded on $CuWO_4$. This scheme illustrates that Co-Pi effectively passivates the surface state of $CuWO_4$ to improve hole transfer efficiency and increase photovoltage [26,70]. S. Chen et al. [52] reported that Co-Pi could not induce a significant negative shift in the onset potential, but Co-Pi/$CuWO_4$ exhibited a higher photocurrent density than bare $CuWO_4$ by 86% at 1.23 $V_{RHE}$ in 0.1 M KPi, as shown in Figure 9b. EIS data of bare $CuWO_4$ at 0.3 $V_{RHE}$ were fitted into an equivalent circuit model comprising a single RC circuit related to the semiconductor–electrolyte interface (Figure 9c). In contrast, EIS data of Co-Pi/$CuWO_4$ aligned closely with an equivalent circuit model featuring two RC circuits. These circuits can be associated with the electron

transport between $CuWO_4$ and CoPi layers, as well as at the semiconductor–electrolyte interface. As a result, the Co-Pi/$CuWO_4$ appears to enhance electron transfer both at the surface of $CuWO_4$ and within the electrode. The Co-Pi was introduced via various methods, such as cyclic voltammetry (CV) [53] and photo-assisted electrodeposition (PED) [26], and both methods formed an amorphous coating layer on a $CuWO_4$ particle. In addition, the Co-Pi/$CuWO_4$ was more stable in electrolytes than bare $CuWO_4$ because the Co-Pi layer could cover the defects of the electrode surface and inhibit the charge accumulation on the surface.

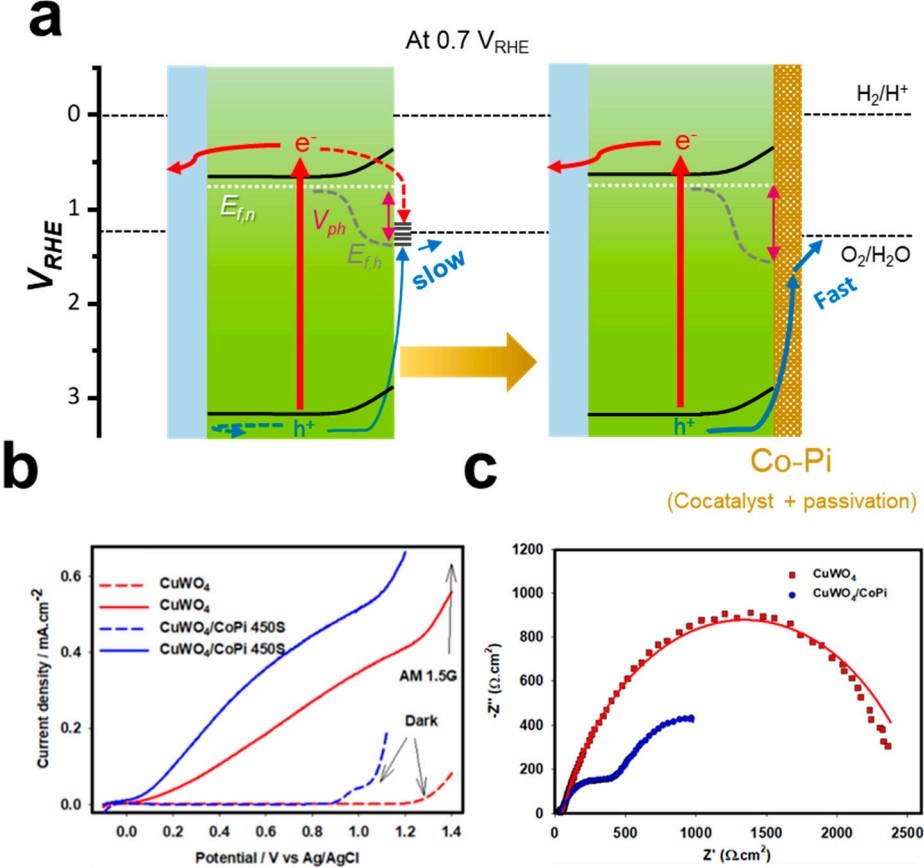

**Figure 9.** Efficacy of Co-Pi for $CuWO_4$ photoanode. (**a**) Working principle of $CuWO_4$ photoanodes with and without a Co-Pi co-catalyst. (**b**) IV curves and (**c**) Nyquist plots at an applied bias of 1.23 $V_{RHE}$ in 0.1 M KPi, pH 7.0 under simulated one sun. Red and blue solid line are for from equivalent circuit. Reproduced from ref. [52].

$Co_3O_4$ was introduced by C. M. Tian et al. [35] as an OER co-catalyst that increased the surface charge separation efficiency by reducing the onset potential of $CuWO_4$ by ~100 mV. Through LSV and EIS analyses, however, they showed that too-thick co-catalyst layer created recombination sites where holes could not pass through, consequently leading to a decrease in PEC performance. The $CuWO_4$ can make the p-n junction with a p-type sulfide electrocatalyst [44]. This junction induced the photogenerated electron and hole transfer to the $CuWO_4$ bulk and p-type sulfide electrocatalyst, respectively. The photocurrent of the $CuWO_4$/p-type sulfide increased and effectively reduced the onset potential by suppressing charge recombination. K. M. Nam et al. [55] compared $Mn_2O_3$ and MnPO co-catalysts in terms of enhancing the efficiency of a $CuWO_4$ photoanode. The OER catalytic performance of MnPO was relatively higher when it was applied to $CuWO_4$/$WO_3$ heterojunction. Unfortunately, the onset potential remained the same, but the stability and efficiency of MnPO/$CuWO_4$/$WO_3$ increased to a reasonable extent. The $NiWO_4$ has a distorted wolframite crystal structure similar to $CuWO_4$ and can form a type

II heterojunction with $CuWO_4$ due to its higher energy CB and VB levels than those of $CuWO_4$, reducing electron–hole recombination and improving IPCE [56]. P. Shadabipour et al. [57] deposited a well-known Ni- and Fe-based co-catalyst on $CuWO_4$, but the OER efficiency of $Ni_{0.75}Fe_{0.25}O_y/CuWO_4$ did not significantly increase because photoexcited holes did not migrate efficiently and the reaction occurred preferentially on the $CuWO_4$ surface rather than on the co-catalyst.

The Ni-Pi synthesized on $CuWO_4$ via a drop-casting method dramatically increased the stability of $CuWO_4$ by interfering with charge accumulation [59]. In addition, it reduced charge recombination to greatly increase PEC performance. Moreover, the authors calculated the decay time of the charges in $CuWO_4$ and Ni-Pi/$CuWO_4$ through a time-resolved photoluminescence (TRPL)-based approach and found that Ni-Pi/$CuWO_4$ had a decay time of 400 μs, which was five times longer than that of bare $CuWO_4$. M. Davi et al. [58] fabricated a MnNCN/$CuWO_4$ electrode through ED and drop-casting. Interestingly, the surface of MnNCN converted to $MnPO_x$, forming a core–shell structure during water oxidation. However, the electrode showed a relatively small increase in activity, and the onset potential did not change. Noble metals were also introduced as co-catalysts for $CuWO_4$. R. Salimi et al. [60] synthesized Ag-functionalized $CuWO_4/WO_3$ using a polyvinyl pyrrolidone (PVP)-assisted sol–gel (PSG) method. This photoanode showed a photocurrent of ~0.2 mA/cm$^2$ at 1.23 $V_{RHE}$ in 0.1 M phosphate buffer due to Ag functionalization, which enhanced charge transfer and separation efficiency and mitigated charge recombination. An IrCo-Pi co-catalyst was deposited on $CuWO_4$ NFs via the ED method to mitigate interfacial electron–hole recombination. The IrCo(9:1)-Pi/$CuWO_4$ exhibited a photocurrent of ~0.54 mA/cm$^2$ at 1.23 $V_{RHE}$. Moreover, the co-catalyst-modified photoelectrode was more robust than bare $CuWO_4$ [61]. In conclusion, the strategy of co-catalyst has been studied in a variety of ways and is still being actively explored. Unfortunately, despite these research efforts, an optimized co-catalyst material that gives outstanding performance has not yet been found, which requires continued research.

## 6. Conclusions

$CuWO_4$ possesses a suitable band gap (2.3 eV) for solar water splitting, which positions it as a promising photoanode with a theoretical STH conversion efficiency exceeding 10%. Since its utilization as a photoanode began in 2011, the performance of $CuWO_4$ photoanodes has been steadily improved through advanced synthesis methods and modification strategies. The most noticeable event in the development of $CuWO_4$ photoanodes was the formation of a solid solution with Mo, thereby reducing the band gap to ~2.1 eV, which resulted in a significant enhancement in solar light harvesting and water oxidation efficiency. In addition, modification strategies proven effective for other well-developed photoelectrodes have been applied to the $CuWO_4$ photoanodes. For example, forming a heterojunction with $WO_3$ or $SnO_2$ provides an ETL for effective charge transfer. Meanwhile, doping an $M^{n+}$ ($n \geq 3$) ion into the Cu octahedral site or using gas treatment to generate oxygen vacancies increases the bulk charge-carrier density. Lastly, co-catalyst loading on $CuWO_4$ offers multiple benefits, such as increasing photovoltage, improving surface hole transfer kinetics, and enhancing chemical stability. A fundamental understanding of the $CuWO_4$ material and its working mechanism under PEC reaction conditions attained via DFT calculations and spectroscopy has facilitated these technical developments.

Despite these steady advancements, the performance of the $CuWO_4$ photoanodes still lags far behind the theoretical expectation. This shortfall comes from intrinsically undesirable PEC properties of $CuWO_4$, like low charge separation efficiency, short hole diffusion length, sluggish surface kinetics, and low bulk conductivity. This signifies the pressing need for the development of an optimal synthesis method and effective modification strategies for $CuWO_4$ in the future. Additionally, strategies used to maximize efficiency by combining $CuWO_4$ with another photoanode with a smaller band gap or integrating it with a high-efficiency photocathode for overall water-splitting systems could also be pursued. The enhancement of $CuWO_4$ and its relevant variant will greatly enlarge

research and application opportunities (e.g., the commercialization of PEC solar overall water splitting) for solar energy utilization technology, just as $BiVO_4$ and $Fe_2O_3$ did for this field. For instance, future research, such as the valorization of organic molecular via partial oxidation, can be conducted using $CuWO_4$ and other tungstates, which will have different and interesting chemistry compared to those of other materials. Also, detailed analysis of the optoelectronic property of $CuWO_4$ will provide much information that is useful for solving the aforementioned problems realized in the current state of development. After this problem has been dealt with, $CuWO_4$ will surely be able to take the place of $BiVO_4$ due to its similarity in terms of operating conditions (near neutral) and elemental composition (ternary metal oxide), which will benefit the community studying photoelectrochemical cells in the same way as previous go-to materials. More specifically, owing to its appropriate band gap that can achieve STH above 10%, it can be possible candidate for a practical solar overall water-splitting PEC cell.

**Author Contributions:** Conceptualization, J.U.L. and J.H.K.; investigation, J.U.L.; data curation, J.H.K.; writing—original draft preparation, J.U.L. and J.H.K.; writing—review and editing, J.S.L.; supervision, J.H.K. and J.S.L.; project administration, J.H.K.; funding acquisition, J.S.L. All authors have read and agreed to the published version of the manuscript.

**Funding:** This work was supported by the Climate Change Response Project (NRF2019M1A2A2065612) and the Brainlink Project (NRF2022H1D3A3A01081140), funded by the Ministry of Science and ICT of Korea via National Research Foundation, and by research funds from Hanhwa Solutions Chemicals (2.220990.01) and UNIST (1.190013.01). This work was also supported by the Institute for Basic Science (IBSR019-D1). J.H.K. gratefully acknowledges the support from the Basic Science Research Program funded by the Ministry of Education (NRF2021R1A6A3A14039651, NRF-2019R1A6A3A01096197).

**Data Availability Statement:** Not applicable.

**Conflicts of Interest:** The authors declare no conflict of interest.

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
