# Peer review of "Emergent CuWO4 Photoanodes for Solar Fuel Production: Recent Progress and Perspectives"

_catalysts, doi:10.3390/catal13111408_

Round 1

Reviewer 1 Report

Comments and Suggestions for Authors

The authors presented a review on the topic "CuWO4 Photoanodes for Solar Fuel Production". The topic selected by them is interesting, and the presentation of data is appropriate. 
The manuscript can be accepted in its present form. 

Author Response

Reviewer #1
Comments to the Author
The authors presented a review on the topic "CuWO4 Photoanodes for Solar Fuel Production". The topic selected by them is interesting, and the presentation of data is appropriate.

The manuscript can be accepted in its present form.

Reviewer 2 Report

Comments and Suggestions for Authors

The review was systematically written.  

For the detailed comments:

1) What are the broader implications of the study's conclusions?

2) Based on the findings, are there any recommendations for future research or practical applications?

3) cite the tungstate-based materials in the introduction. Environmental Research 228 (2023) 115851. Inorganics 11 (2023) 213.

Author Response

Reviewer #2
Comments to the Author
The review was systematically written. 

For the detailed comments:

Comments 1:

What are the broader implications of the study's conclusions?

Response 1: It is a good point to consider broader implications of this review, since our current conclusion was written in mostly just scientific discovery, no implications handled. Both of comment 1 and 2 are reflected to conclusion by adding more line into it.

Added

Page 18, line 686

“Enhancement of CuWO4 and its relevant variant will greatly enlarge research and ap-plication opportunity (e.g. commercialization of PEC solar overall water splitting) for solar energy utilization technology, just as BiVO4, Fe2O3 did for this field. For instance, future research such as valorization of organic molecular by partial oxidation can be done by CuWO4 and other tungstate’s, which will have different and interesting chemistry than what they are for other materials. Also, detailed analysis for optoelectronic property of CuWO4 will provide many of information which is useful for solving aforementioned problems realized in current state of development. After problem is being dealt with, CuWO4 will surely be able to substitute the place of BiVO4 for its similarity of operating condition (near neutral) and elemental composition (ternary metal ox-ide), which shall benefit community studying photoelectrochemical cell once again as previous go-to materials. More specifically, owing to its appropriate band gap that can achieve STH above 10 %, this can be possible candidate for practical solar overall water splitting PEC cell.”

Comments 2:

Based on the findings, are there any recommendations for future research or practical applications?

Response 2: It is a good point to mark future research and possible practical application for CuWO4. Yet since it is one of the materials for photoelectrochemical water oxidation, its potential aspect will be more or less the same with other photoanodes, but with dependence on its chemistry (being tungstate). We can specifically note that study of charge carrier dynamics for deeper understanding of CuWO4 as optoelectronic material, as well as partial oxidation of various chemicals for valorization reaction via photoelectrochemical cell for future research, while reaching high efficiency around 10 % of STH will enable practical application. It is also reflected in conclusion as noted from comment 1.

Comments 3:

Cite the tungstate-based materials in the introduction. Environmental Research 228 (2023) 115851. Inorganics 11 (2023) 213.

Response 3: We thank reviewers for introducing relevant literatures, we reflected those to introduction of our manuscript.

Page 2, line 85

Added

“which is amongst photoactive ternary tungstates (Bi2WO6 which showed PEC water oxidation activity[14], ZnWO4 which showed pollutant degradation [15]and PEC water oxidation activity [16])”

Added references

  1. Hill, J.C.; Choi, K.-S. Synthesis and characterization of high surface area CuWO4 and Bi2WO6 electrodes for use as photoanodes for solar water oxidation. J. Mater. Chem. A 2013, 1, 5006-5014.
  2. Bathula, B.; Eadi, S.B.; Lee, H.-D.; Yoo, K. ZnWO4 nanorod-colloidal SnO2 quantum dots core@shell heterostructures: Efficient solar-light-driven photocatalytic degradation of tetracycline. Environmental Research 2023, 228, 115851.
  3. Babu, B.; Peera, S.G.; Yoo, K. Fabrication of ZnWO4-SnO2 Core-Shell Nanorods for Enhanced Solar Light-Driven Photoelectrochemical Performance. Inorganics 2023, 11.

Reviewer 3 Report

Comments and Suggestions for Authors

Reviewer comments

This manuscript wants to present a review on CuWO4 photoanode for solar fuel production. It briefly described the principal features of CuWO4 and gives an overview of important examples of recent approaches to improve the water splitting efficiency. The manuscript is meaningful and well written, however, there are some issues should be addressed before it is accepted by Catalysts.

1. The authors need to give more details of CuWO4 photoanode in the introduction part. It is true that not much research output has been done on CuWO4 compared to the BiVO4 photoanode mentioned by the author. We hope the authors can explain the reasons and describe the manuscript with ways to improve them.

2. From the comments 1, this seems to be the reason why there are few recent papers for the studies mentioned in this manuscript. If there are more recent literatures, since 2020, please cite them.

3. It would be better if characterization results such as XRD and Sem were also shown in Figure 2.1, which explains the CuWO4 materials.

4. Proof of stability is one of the important factors in the materials, however the results in KBi or low pH solutions mentioned in session 3.2 confuse us. This is because photoanode experiments are commonly used in neutral pH or basic environments.

Author Response

Reviewer #3
Comments to the Author
This manuscript wants to present a review on CuWO4 photoanode for solar fuel production. It briefly described the principal features of CuWO4 and gives an overview of important examples of recent approaches to improve the water splitting efficiency. The manuscript is meaningful and well written, however, there are some issues should be addressed before it is accepted by Catalysts.

Comments 1:

The authors need to give more details of CuWO4 photoanode in the introduction part. It is true that not much research output has been done on CuWO4 compared to the BiVO4 photoanode mentioned by the author. We hope the authors can explain the reasons and describe the manuscript with ways to improve them.

 Response 1: We agree about reviewer #3’s concern about short details handled in introduction for CuWO4. However, since this review dedicates to CuWO4, all the aspects we studied for this material are handled in later sections (Crystal structure, electronic structure, synthesis, physiology being photocatalyst and photoelectrode and modification) of this manuscript. We would like to request reviewer to consider this point.

Comments 2:

From the comments 1, this seems to be the reason why there are few recent papers for the studies mentioned in this manuscript. If there are more recent literatures, since 2020, please cite them.

Response 2: We are afraid that we could not find any more of recent manuscripts related to CuWO4. For instance, here is a list of literatures (published in 2023) we handled.

  1. Lee, J.U.; Kim, J.H.; Kang, K.; Shin, Y.S.; Kim, J.Y.; Kim, J.H.; Lee, J.S. Bulk and surface modified polycrystalline CuWO4 films for photoelectrochemical water oxidation. Renewable Energy 2023, 203, 779-787.
  2. González-Poggini, S.; Sánchez, B.; Colet-Lagrille, M. Enhanced Photoelectrochemical Activity of CuWO4 Photoanode by Yttrium Doping. Journal of The Electrochemical Society 2023, 170, 066512.
  3. Ikeue, K.; Ueno, T. Photoelectrochemical water oxidation properties of Mo-doped CuWO4: Effect of p-type sulfide loading and annealing. Materials Letters 2023, 348, 134690.
  4. Wang, K.; Chen, L.; Liu, X.; Li, J.; Liu, Y.; Liu, M.; Qiu, X.; Li, W. Gradient surficial forward Ni and interior reversed Mo-doped CuWO4 films for enhanced photoelectrochemical water splitting. Chemical Engineering Journal 2023, 471, 144730.

Comments 3:

It would be better if characterization results such as XRD and Sem were also shown in Figure 2.1, which explains the CuWO4 materials.

Response 3: We added an XRD data set for Figure 2, which we believe it is helpful to show more information about the material we handled.

Figure 2. Solid state reaction of WO3 into CuWO4 thin film. (a) Schematic illustra-tion of the preparation procedure of the CuWO4 NF array film on an FTO substrate. Optical images represent WO3 film, WO3 film drop-cast with Cu(NO3)2 solution, a mix-ture film of CuWO4 and CuO, and CuWO4 film. (b) XRD pattern of CuWO4 film an-nealed at 450 °C, 550 °C and 650 °C. (c) SEM images of the WO3 and CuWO4 nano-flakes prepared at 550 °C for 2 hr and (d) IV curve of CuWO4 nano-flake photoanode (0.1M NaBi buffer, pH 9.0) under simulated 1 sun. Reproduced from ref.[24]

Page 5, right column, line 2

Added

“The X-ray diffraction (XRD) analysis initially showed a distinct monoclinic WO3 pattern in a film, with specific diffraction peaks at 23.3° and 24.5° as shown in Figure 2b. However, following a solid-phase reaction at 550°C for 2 hours and the elimination of excess CuO at the surface, the WO3 pattern disappeared completely, replaced by a clear XRD pattern of triclinic CuWO4. This suggests a complete transformation of WO3 into CuWO4 at the lower temperature of 550°C.”

Comments 4:

Proof of stability is one of the important factors in the materials, however the results in KBi or low pH solutions mentioned in session 3.2 confuse us. This is because photoanode experiments are commonly used in neutral pH or basic environments.

Response 4: We believe that the literature [33] tried to compare WO3 and CuWO4, while as they noted as well as us in our review paper, WO3 has inherent stability in pH around 1~4 instead of pH 7 while CuWO4 showed inherent stability in pH round 7 to 9, which was the difference we wanted to describe in this review paper.
